# Fatigue Life Improvement of Weld Beads with Overlap Defects Using Ultrasonic Peening

**DOI:** 10.3390/ma16010463

**Published:** 2023-01-03

**Authors:** Seung-Hyon Song, Chang-Soon Lee, Tae-Hwan Lim, Auezhan Amanov, In-Sik Cho

**Affiliations:** 1Department of Advanced Materials Engineering, Sun Moon University, Asan 31460, Republic of Korea; 2Department of Mechanical Engineering, Sun Moon University, Asan 31460, Republic of Korea

**Keywords:** welding technology, overlap defect, ultrasonic peening, residual stresses, repair technologies

## Abstract

Welding defects are common during the production of large welded structures. However, few studies have explored methods of compensating for clear welding defects without resorting to re-welding. Here, an ultrasonic peening method to compensate for the deteriorated mechanical properties of overlap weld defects without repair welding was studied. We experimentally investigated changes in the mechanical properties of defective welds before and after ultrasonic peening. The weld specimen with an overlap defect contained a large cavity-type defect inside the weld bead, which significantly reduced the fatigue life. When the surface of the defective test piece was peened, the fatigue life of the weld plate was restored, resulting in an equivalent or higher number of cycles to failure, compared to a specimen with a normal weld. The recovery of mechanical properties was attributed to the effect of surface work hardening by ultrasonic peening and the change in stress distribution. Thus, ultrasonic peening could compensate for the deterioration of mechanical properties such as tensile strength, fatigue life, and elongation due to overlap defects, without resorting to repair welding.

## 1. Introduction

Welding defects are almost inevitable during the production of large welded structures such as ships. They are particularly likely to occur during manual arc welding, as the weld integrity is highly dependent on the skill and condition of the welder on the given day. Owing to the extremely bright light generated during welding, the welder cannot clearly observe the movement of the welding rod, which affects the welding quality based on the welder’s skill. Additionally, improper movement of the welding electrode can lead to undercut and overlap defects of the weld bead [1,2,3].

Welding defects are classified as internal or external. Internal defects, including hydrogen embrittlement, cracks generated by low-temperature embrittlement, porosity, poor penetration, and internal cracks, can only be detected during quality assurance checks such as non-destructive testing. Conversely, external defects, such as undercuts, overlaps, discontinuities, spatter, and slag inclusions, can be identified by visual checks during and after welding. Depending on the severity of the defect, it may require repair welding. Repair welding is the process of digging out and re-welding defective parts. For example, external defects can significantly degrade the mechanical properties of welded structures [4,5,6]. In structural components, this can result in large-scale collapse of the structure, which has the potential to cause severe accidents, injury, and even loss of life. Therefore, a strict inspection process is conducted with non-destructive testing, and welds that do not meet the required standard are unconditionally removed and welded again.

However, repair welding carries certain risks and can introduce fatal defects. For example, the defective part must be removed by using a grinder or cutter to dig out the welded part. During this process, frictional heat is generated, and this can cause phase transformations or grain coarsening in the surrounding material. Further heat is generated during re-welding, which can also cause material deformation and grain coarsening. Coarse grains in the heat-affected zone (HAZ) can grow even larger. Ultimately, these effects increase the risk of reheat cracking and degrade the mechanical properties such as toughness. Furthermore, correcting all minor defects in non-critical parts would incur high process costs. Therefore, for low-severity or “ambiguous” defects, such as defects in parts that receive little load, there is often a debate between the construction and supervisory parties as to whether the defects should be repaired. To satisfy both parties while reducing the risks and costs associated with repair welding, it may be possible to compensate for the performance degradation caused by defects without resorting to repair welding. However, few studies have explored methods of compensating for clear welding defects without resorting to re-welding [7,8].

Peening is a surface treatment technique in which the surface of the material is microplastically deformed [9,10]. This compresses surface pores, hair cracks, and subsurface pores; work-hardens the surface and subsurface of the material; and applies compressive residual stress to the surface. Consequently, the surface hardness and strength of the material are increased, the notch effect is reduced, and the material’s fatigue life is prolonged. Furthermore, the compressive residual stresses limit crack growth at the surface, which improves the material’s resistance to stress corrosion cracking, stress fatigue cracking, and fatigue corrosion [11,12]. Thus, peening is a useful method of improving the fatigue life of components. Many researchers have studied the fatigue life of welded structures subjected to cyclic loads, as well as the changes in fatigue life and mechanical properties after surface treatments such as laser and ultrasonic peening [13,14,15,16,17].

There are several types of peening techniques. Among these, ultrasonic peening uses ultrasonic resonance to impact the surface of a material using needles or impact pins at a rate of 2000 times (20 kHz) to microplastically deform the surface. A major advantage of ultrasonic peening is its portability, which facilitates its on-site application. In addition, by adjusting the quantity and size of the impact tips, peening can be achieved over large areas in a short time [18]. Peening has been shown to change the fatigue life or mechanical properties of welds. However, most studies have focused on normal welds. In this study, we tested whether overlap defects, which are common external weld defects, can be corrected by ultrasonic peening, without resorting to repair welding. We experimentally investigated changes in the mechanical properties of defective welds after ultrasonic peening.

## 2. Materials and Methods

### 2.1. Materials

Welding was performed on hot-rolled carbon steel sheets purchased from Chungnam Steel Co., Ltd., Seoul, Republic of Korea. The grade used was ASTM A570 Gr.40, which is widely used in structures in the shipbuilding and automotive industries. It is considered a low-carbon steel as its carbon content is below 0.25 wt.%. The higher the carbon content, the higher the tendency for martensite to form on cooling; therefore, precautions such as pre-heating or post-weld heat treatment are necessary for welding [19]. Typically, hot-rolled carbon steels such as that used in this study do not require pre- or post-heating for welding. Butt-welded test specimens were fabricated from the cut sheets of rolled plates. Schematics of the butt-welded specimens are shown in Figure 1.

Flux-cored arc welding was performed at a welding current of 280 A and voltage of 30 V. Two types of test plates were produced. The first was manufactured assuming a normal welded product without external weld defects, whereas the second was manufactured to simulate overlap defects that occur during welding. In general, welding that reproduces defects is more difficult than normal welding, so a highly skilled welder worked to reproduce the overlap defects. Welding was performed in an indoor workplace (temperature: 22 °C) without external air flow. Before welding, the lower ends of the plates were first joined by tack welds. We used flux-cored wire welding rods of high-tensile steel based on the AWS A5.20 E71T-1C standard. The welding rods were KISWELL, SF-71. Figure 1c,d show photographs of the welded plates with a normal weld and overlap defect, respectively. The overlap phenomenon in Figure 1d shows that the weld beads have an upward bias [20,21,22,23,24].

Ultrasonic peening was performed on the surface of the weld bead and the HAZ of the welded plate with the overlap defect. The area showing surface discoloration caused by the heat of welding was set as the HAZ. The peening treatment was applied to half of the entire welded portion of the plate with the overlap defect; the other half was not peened for comparing the properties before and after peening. Figure 2a,b depict a schematic and photograph of the ultrasonic peening device, respectively. Ultrasonic peening of the weld surface with overlap defects was performed with a displacement of 20 µm for 120 s. Three impact pins with 6 mm round tips were used to facilitate microplastic deformation. During ultrasonic peening, the peening device was directly picked up by the experimenter to treat the surface. Moderate pressure was applied to the surface along with the weight of the peening device, and no strong mechanical force was applied.

Fatigue and tensile test pieces were fabricated from the welded parts via wire electric discharge machining. To understand the effect of peening, the surfaces containing the weld bead and HAZ were not processed. The shape of the test piece was designed to ensure that the joint of the root part was not retained on the test piece. Additionally, the surface opposite the weld was shaped to create a gauge region, thereby minimizing the cross-sectional area of the weld and HAZ to ensure that fracture occurred in that region. Figure 3 shows a schematic of the fabricated specimen.

The specimens were divided into three types, depending on the weld type and surface treatment. The first type contained the normal weld; the second type was obtained from the defective plate with the overlap defect without peening; and the third type was prepared from the overlap-defect area subject to ultrasonic peening treatment. To observe the microstructure of the welds and measure the microhardness, specimens were prepared by removing the ends of the fatigue test pieces.

### 2.2. Test Procedures

The surfaces, cross-sections, and microstructures of the weld beads and HAZs of the overlap-defect specimens with and without peening were observed using an optical microscope (OM; MF-A1010D, Mitutoyo, Japan). The microstructures of the surfaces were further examined by scanning electron microscopy (SEM; MAIA3, Tescan, Czech Republic). The microhardnesses of the peened and untreated portions of the overlap welded specimen were measured at loads of 2 N using a Vickers hardness tester (Micromet3, Buehler, Lake Bluff, IL, USA). The microhardnesses were measured (see Figure 4a) in the HAZ on the left (point A) and right (point B) sides of the weld bead and in the weld bead itself (point C). The measurements were performed at 0.2–0.4 mm intervals on the cross-sections of the samples, starting at a depth of 0.2 mm from the surface.

Tensile and fatigue tests were performed on the normal welded specimens, specimens with the overlap defects, and specimens with peened overlap defects using a universal testing machine (MTS, 810). Tensile tests were performed at a tensile speed of 1 mm/min to measure the tensile and yield strengths and obtain stress–strain curves. In the fatigue tests, stresses of +170 and −170 MPa were applied to the specimens alternately at a frequency of 20 Hz. The fatigue life was calculated as the average number of cycles to failure of five tests. The 170 MPa load was used as it is approximately 65% of the yield strength. The fatigue fracture surfaces were observed using SEM (JSM-7500F, JEOL Ltd., Tokyo, Japan).

Finally, the residual stress of the cross-section was measured using an X-ray residual stress meter (SmartLab 9KW, Rigaku, Japan) with a Cu Kα X-ray source, power of 45 kV, current of 200 mA, *θ*/2*θ* scan mode, and 2*θ* scan range of 10.0–90.0°. Three points were measured per specimen to measure the residual stress distribution (see Figure 4b): the top of the HAZ on the specimen (point A), the top of the bead (point B), and the bottom of the bead (point C). The purpose of this residual stress measurement was to identify the change in residual stress distribution when an overlap defect occurred and the change in residual stress when the surface was peened.

## 3. Results

### 3.1. Surface Morphology of Overlap-Defect Weld

Figure 5a shows a photograph of the defective plate after peening. Peening was performed on half of the welded section. Numerous small dimples were formed on the peened surface. The enlarged image in Figure 5b shows that the weld bead was compressed during the peening process. In particular, the overlap region exhibited a large amount of compression. Although the compression was not substantial on the left side of the weld bead (see Figure 5b), a significant amount of compression was generated by ultrasonic peening on the right side of the overlapping section. This indicates that numerous unstable defects existed inside the overlapping section.

### 3.2. Microstructure

The HAZ and bead part of the welded specimen with the overlap defect without peening were observed using OM. The cross-section of the bead exhibited a typical weld dendrite microstructure, as shown in Figure 6a. Figure 6b depicts the microstructure of the HAZ, wherein coarsened pearlite was observed on the right side owing to the welding heat, and the ferrite–pearlite structure of the base material was observed on the left side. The dark and light regions are pearlite and ferrite, respectively. Figure 6c depicts a magnified image of the upper part of the HAZ, demonstrating that there was a decarburized layer on the upper surface. The partial progression of surface decarburization was attributed to the heat transferred from the bead during welding. It is important to avoid surface decarburization because it reduces fatigue life [25,26].

The specimen with the peened overlap defect was observed under OM. The compression effect caused by peening was observed on the HAZ surface. Figure 7a,b show low- and high-magnification images, respectively, of the compressed layer on the surface of the HAZ after ultrasonic peening. The high-magnification image confirmed the compression based on the shape of the ferrite and pearlite grains. These changes were attributed to the effect of surface compression caused by the local plastic deformation during peening. The SEM image in Figure 7c clearly shows the surface compression layer of the HAZ area caused by ultrasonic peening. This layer was approximately 10 µm thick, with the grain boundaries disappearing due to the compression. Figure 7d shows a high-magnification SEM image of the surface of the weld bead, with no significant change observed in the microstructure because of the compression effect. In addition, it appears that the decarburized layer at the top of the HAZ before peening was compressed. When decarburization occurs, carbon in Fe_3_C in the pearlite structure escapes, leaving a soft ferrite structure. By peening, the decarburized layer was work-hardened and turned into a compressed layer on the surface.

### 3.3. Microhardness

The microhardness was measured from the surface of the section to a 3.0 mm depth. The measurement positions were the center of the bead and the left and right sides of the HAZ. Table 1 summarizes the measurement results. For the overlap-defect specimen without peening, the average hardness value from the surface to the 3.0 mm-deep point of the weld bead was 160.9 HV; no significant deviation was observed with respect to depth. Additionally, the average hardness values of the HAZ on the left and right sides of the weld bead were 123.7 and 126.6 HV, respectively. Again, no significant deviation was observed with respect to depth.

For the ultrasonically peened overlap-defect specimen, the average hardness of the HAZ varied significantly with depth. Between 0.2 and 0.8 mm, the average hardness values of the HAZ on the left and right sides of the weld bead were 169.5 and 170.4 HV, respectively, which corresponds to a 40% improvement compared to the average hardness values before peening. This improvement in hardness was attributed to the work-hardening effect caused by ultrasonic peening. Subsequently, the hardness gradually decreased with an increase in depth, reaching approximately 120 HV at a depth of 3.0 mm. The rate of increase in hardness of the bead part owing to peening was lower than that of the HAZ; nevertheless, the hardness also increased in the bead part because of work hardening. The average hardness of the peened weld bead from the surface to a depth of 0.8 mm was 190.6 HV, which corresponds to an increase of approximately 15% compared to that before peening (160.9 HV). The improvement of surface hardness was also confirmed in the bead part. The hardness of the bead part was 172.6 HV at a depth of 1.0 mm, which was greater than that at the same depth before peening (165.7 HV).

### 3.4. Tensile Properties

Table 2 summarizes the tensile test results. For the normal welded specimen, the tensile and yield strengths were 399 and 277 MPa, respectively. Thus, this specimen satisfies the 400 MPa strength standard of A570 Gr.40. The strength softening phenomenon caused by welding was insignificant because the area of grain coarsening in the HAZ was relatively small. In the case of the specimen with overlapping defects, the yield strength was 231 MPa, which was lower than that of the normal welded specimen (277 MPa); however, the tensile strength was 345 MPa, which is 20% lower than that of the normal welded specimen. The normal welded specimen fractured in the HAZ during the tensile test, which is typical for welded specimens, whereas the fracture was initiated in the weld bead in the overlap-defect specimen. This caused rapid failure of the specimen owing to the concentration of stress on the defects inside the overlap. Furthermore, the elongation of the overlap-defect specimen subjected to the tensile test was reduced to 17.8%, which is 33% less than that of the normal welded specimen (26.5%).

For the peened overlap-defect specimen, both the tensile and yield strengths increased in comparison with those of the normal welded and overlap-defect specimens. The measured yield and tensile strengths were 289 and 434 MPa, respectively. The tensile strength was increased by peening because the overlap defect was largely compensated for by peening. The improvement in strength was attributed to two aspects: first, residual stress was imparted to the test piece during the peening operation; second, the work-hardening effect improved the surface and subsurface strengths, because the Vickers hardness test confirmed that the hardness increased due to peening, and the strength of a metallic material generally improves with an increase in hardness. Moreover, work hardening decreases the elongation and ductility of welded materials. Although the elongation of the peened overlap-defect specimen was improved compared to that of the overlap-defect specimen, the elongation was decreased in comparison with that of the normal welded specimen. Figure 8 shows the stress–strain curves of the normal welded, overlap-defect, and overlap-peened specimens. When low-carbon steel is press-processed, a yield point drop phenomenon can occur, which causes surface wrinkles called stretcher strains. However, these wrinkles were not observed on the surface of the peened specimen. This was because the strength after peening was greater than the yield point [27].

### 3.5. Fatigue Behavior

Table 3 summarizes the results of the five fatigue tests for each type of specimen. The average fatigue cycle life of the normal welded specimens was 111,725 cycles. The fracture surfaces were observed by SEM to elucidate the fatigue failure mode, as shown in Figure 9. The low-magnification SEM image in Figure 9a shows that the main failure mode was brittle fracture. At higher magnification (see Figure 9b), evenly distributed fatigue striations were observed over most of the fracture surface. Figure 9c shows the morphology of the final fracture at the end of the test. Brittle regions and ductile dimpling were sequentially arranged. Additionally, the fracture occurred in the HAZ, as indicated by the tensile test results. Combining these factors, we concluded that the fracture of the normal welded specimen was typical of welded specimens under fatigue cycling.

For the overlap-defect test pieces, the average fatigue cycle life was 2423, which is extremely low compared to that of the normal weld specimens. Figure 9d shows a low-magnification SEM image of the test piece, wherein a large defect was observed inside the fracture surface. This phenomenon was caused by incomplete filling of the butt weld by filler material, because the bead was biased to one side during welding owing to the nature of the overlap defect. Therefore, the fatigue life of the overlap specimen was significantly reduced compared to that of the normal welded product because of the large defects in the cavity.

The low-magnification SEM image in Figure 9d also depicts that the sample failed by brittle fracture. A brittle fracture region was observed across the entire width of the test piece, indicated by the red arrows. However, no fatigue striations were observed on this large brittle fracture surface. The absence of fatigue striations was attributed to the instantaneous expansion of the defect when the specimen was subjected to fatigue stress. Nevertheless, fatigue striations were observed at the top left of the fracture surface, as marked by red circles in Figure 9e,f. The fatigue crack was initiated in this region. Based on our analysis, we concluded that fatigue failure occurred rapidly because fatigue stress was applied to the narrow cross-sectional area, which resulted in the instantaneous expansion of the defect.

The average fatigue life of the peened overlap-defect specimens was significantly improved to 118,970 cycles. Notably, this is even better than that of the normal welded specimens (111,725 cycles). This improvement was confirmed by visually inspecting the specimens following the fatigue failures. For the overlap-defect specimen, the fracture initiated at the incomplete penetration defect inside the weld bead. Figure 10a shows a photograph of an overlap-defect specimen after the fatigue test; the fracture site was in the center of the weld bead. In contrast, the peened overlap-defect specimen fractured in the HAZ, even though a cavity-shaped defect was present in the bead part at the center of the specimen, as indicated by the white arrow (see Figure 10b). Similarly, the normal welded specimen also fractured in the HAZ.

Figure 10c–h depict the SEM images of the fatigue fracture surfaces of the peened overlap-defect specimens. The fracture surface can be divided into three zones, as shown in Figure 10c. The first (marked with a red arrow at the bottom of the above figure) is the HAZ surface and subsurface area, where the surface was compressed by ultrasonic peening. The second (marked with a yellow circle) is the fatigue striation region, which indicates the progression of the fatigue cracks. Finally, in the third zone (at the top of the above figure), ductile and brittle regions coexisted, with no fatigue striations. Therefore, the third zone was considered to be the final stage of rapid failure.

Figure 10d shows a higher magnification image of the compressed HAZ surface. No ductile fracture or fatigue striations were observed in this area. In comparison with the general brittle section, the fracture surface was smooth in this region. The fracture in this compressed area and the final fracture region are believed to have occurred simultaneously. As shown in Figure 10e, fatigue striations were observed immediately below the compressed zone. These fatigue striations initiated at the compression zone below the HAZ surface and propagated through the specimen. Figure 10f confirms that the fatigue striations progressed entirely in this striation region. In contrast, fatigue striations did not appear in the ductile–brittle fracture area, as shown in Figure 10g. This is because the final fracture occurred after the fatigue crack growth. Examining this area indicated that the brittle and ductile surfaces were composed of fine layers. Figure 10h shows the brittle surface and ductile-type dimple morphology [28,29,30,31,32]. When the final fracture occurred, the ductile layer was presumed to be slightly elongated, delaying the final fracture.

### 3.6. Residual Stress

The residual stresses in the cross-sections of the overlap-defect and peened overlap-defect specimens were measured using an X-ray residual stress measuring device. Initially, X-ray diffraction was performed to select a plane for measuring the residual stress; Figure 11 illustrates the results. Among the X-ray diffraction peaks, the (211) plane was selected for residual stress measurements, which could generate data about all three lattice parameters (*a*, *b*, and *c*). As the residual stresses determine the lattice distortions in the three directions, this plane was considered ideal for the measurements.

In general, the welded section comprises the HAZ, base metal, boundary, and weld bead. The typical stress distribution includes tensile residual stresses in the HAZ and weld bead. During welding, the temperature of the weld bead increases above the melting temperature of the material because of the arc heat, and the bead changes to a liquid phase. Consequently, the volume of the bead expands, followed by contraction as the bead solidifies. In the HAZ, heat is transmitted from the weld bead to the surrounding material, increasing the temperature to the austenite zone. Subsequently, the material rapidly cools, which results in the formation of a martensite or pearlite phase, leading to volume expansion. Thus, the weld bead and HAZ exhibit residual tensile stresses owing to the volume expansion.

Table 4 summarizes the residual stress measurement results. For the overlap-defect specimen, the compressive stress was −386 MPa in the HAZ, with the tensile stresses at the top and bottom of the bead part 1496 and 1037 MPa, respectively. This stress state was attributed to the cavity-type defect observed in the center of the bead of the overlap-defect specimen owing to the incomplete penetration of the filler metal during welding. This cavity creates a strong expansion force on the surrounding material. Therefore, a high residual tensile stress of 1496 MPa was measured at the top of the bead owing to the expansion force; however, it acted as a repulsive force on the surrounding material, leading to a compressive residual stress being measured in the HAZ. This is because the volume of the base material was large in the HAZ. Thus, the rapid failure of this specimen during fatigue testing was exacerbated by amplifying effect of the residual tensile stress in the weld bead on the applied fatigue stress. Figure 12a,b depict this phenomenon schematically.

The residual stress distribution in the overlap-defect welded plate changed after ultrasonic peening. Notably, the respective stresses at the surface and lower parts of the bead decreased from 1496 and 1037 MPa before peening to 460 and 647 MPa after peening. As the surface of the bead was directly peened using the ultrasonic peening device, the decrease in the tensile stress on the surface was large, whereas a smaller decrease was observed in the lower part of the bead. Conversely, the compressive residual stress of −386 MPa in the HAZ changed to a tensile residual stress of +978 MPa after ultrasonic peening. This change in the stress distribution amplified the residual tensile stress in the HAZ during fatigue testing of the peened specimen, which resulted in fatigue failure occurring in the HAZ without significant expansion of the fatal defects inside the bead. Figure 12c schematically illustrates this phenomenon, while Figure 13 and Figure 14 show the respective residual stress diagrams.

## 4. Discussion

To determine the efficacy of ultrasound peening for correcting overlap defects without repair welding, normal and overlap-defect welded specimens were produced and tested. The most prominent feature in this experimental study was the significant reduction in the fatigue life of the welded specimens with overlap defects. The fatigue life of the overlap-defect welded specimens was approximately 50 times lower than that of the normal welded specimens. This was caused by the large cavity defect inside the weld bead, where the filler metal was biased in one direction and thus did not completely infiltrate the weld bead. This large internal defect resulted in rapid crack progression when fatigue stress was applied. The fracture was located in the weld bead itself.

The welded plates with overlap defects were then subjected to ultrasonic peening. Fatigue tests showed that the peened specimen had a slightly higher fatigue life than the normal welded specimen, and a significantly higher fatigue life than the overlap-defect specimen. Furthermore, the fatigue failure occurred in the HAZ in the peened specimen. Despite the presence of defects inside the bead, they were not enlarged under the applied fatigue load, and the fracture occurred in the HAZ, similar to that observed in the normal welded specimen.

The higher fatigue strength of the peened specimen was due to two reasons. The first reason is that the high residual tensile stress in the bead caused by the defects was lowered by ultrasonic peening and converted into a relatively high tensile residual stress in the HAZ rather than the bead. This implied that elongation occurred in the HAZ during fatigue cycling. The second reason is the effect of surface work hardening. The increase of hardness due to work hardening was much higher in the HAZ than in the bead. The higher the surface hardness, the higher the fatigue life. Therefore, elongation during fatigue cycling occurred in the HAZ, and because the surface hardness of the HAZ was increased, the fatigue life increased compared to that of the normal weld. The decrease in tensile strength and elongation due to the overlap defect were therefore compensated for during fatigue cycling. The above results demonstrate that ultrasonic peening technology can compensate for the elongation and reduction in tensile strength and fatigue life caused by the overlap defects of welds without repair welding.

Furthermore, two benefits can be obtained from ultrasonic peening technology. Because inclusions and segregations inside the material also lead to pores and cracks, defect detection is sometimes inconclusive in non-destructive testing. Such defects can be cost-effectively repaired by ultrasonic peening instead of re-welding all potential defects. Furthermore, our findings show that ultrasonic peening could compensate for non-safety-critical overlap defects, such as those in structures subjected to weak stresses (e.g., support rods for long pipes) [33,34,35,36,37,38]. Welding overlap defects in such parts can be easily repaired with ultrasonic peening instead of re-welding. Therefore, minimal re-welding of overlap defects could reduce maintenance costs.

## 5. Conclusions

In this study, ultrasonic peening was explored to compensate for frequent overlap defects during welding. The surface of the welded plate, which reproduced the overlap defect, was treated using ultrasonic peening, and the mechanical properties before and after peening treatment were compared. When an overlap defect occurred, a large cavity defect formed inside the bead. Defects inside the bead lowered the tensile strength and greatly reduced the fatigue life. Nevertheless, both tensile strength and fatigue life were compensated as a result of ultrasonic peening treatment for this overlap defect. Due to surface hardening by ultrasonic peening and change in stress distribution, cracks did not develop inside the overlap bead. Therefore, ultrasonic peening could compensate for the deterioration of mechanical properties such as tensile strength, fatigue life, and elongation due to the overlap defect. However, the change in absorbed energy during impact tests, which is another important mechanical property, should be further studied. In addition, it is essential to set the work standard for utilizing ultrasonic peening in actual weld sites; therefore, the effects of peening for different times and pressures per unit area should be further studied.

## Figures and Tables

**Figure 1 materials-16-00463-f001:**
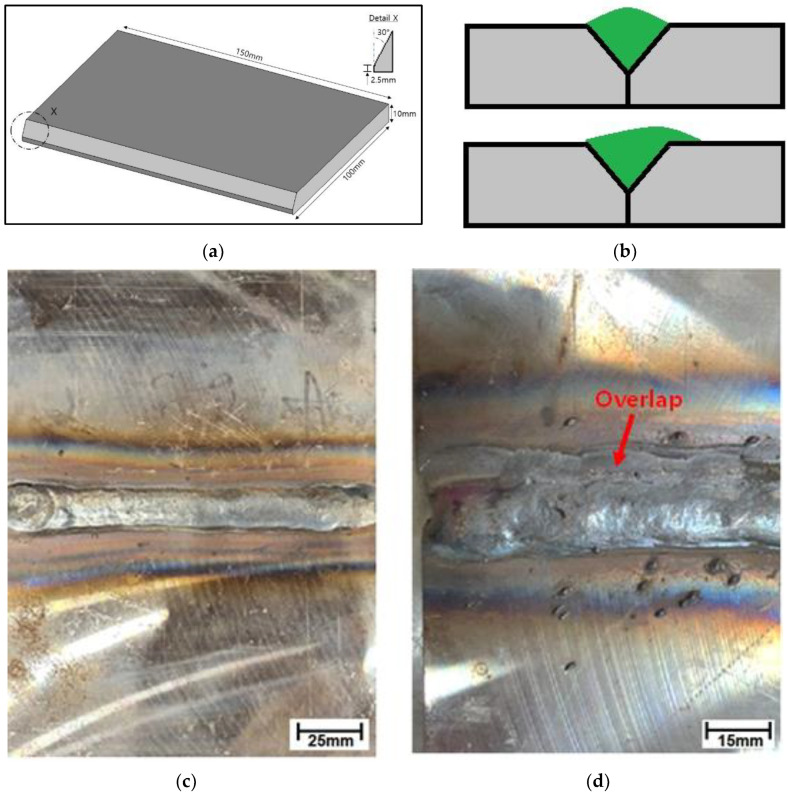
Weld specifications. Schematics of (**a**) cut plate for forming butt welds and (**b**) cross-section of normal weld (**top**) and overlap weld (**bottom**); Photographs of (**c**) normal welded plate and (**d**) welded plate with overlap defect.

**Figure 2 materials-16-00463-f002:**
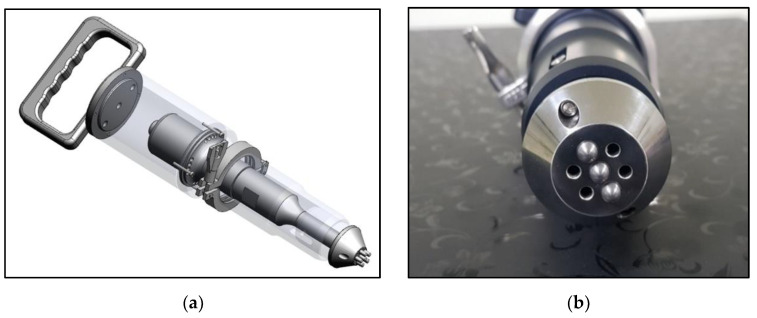
Ultrasonic peening device. (**a**) Diagram of an ultrasonic peening device; (**b**) Photograph of a three-tip ultrasonic peening device.

**Figure 3 materials-16-00463-f003:**
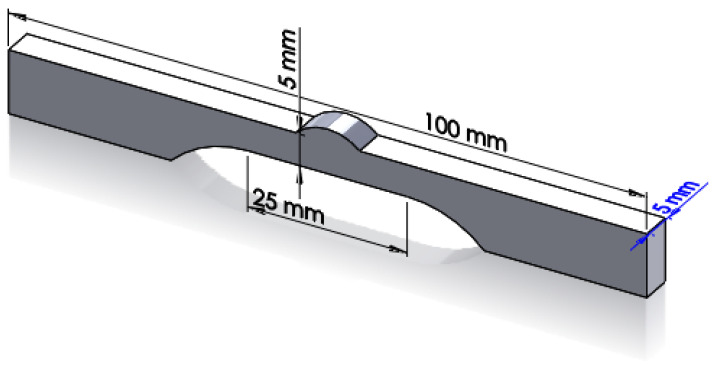
Schematic of the tensile and fatigue test specimen obtained from the welded plates.

**Figure 4 materials-16-00463-f004:**
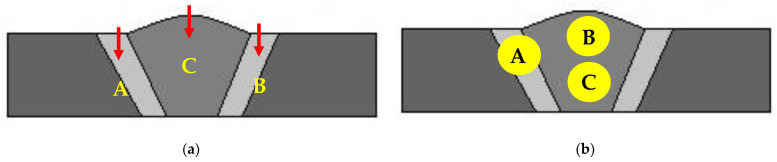
Measurement positions on weld specimens. (**a**) Microhardness measurement positions; (**b**) Residual stress measurement positions.

**Figure 5 materials-16-00463-f005:**
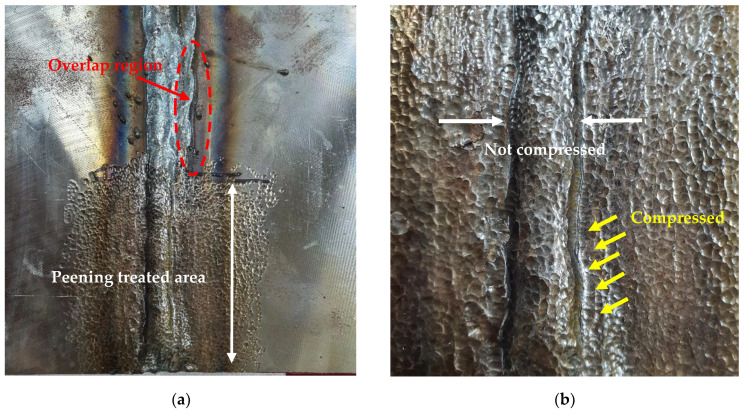
Photographs of overlap-defect weld after ultrasonic peening. (**a**) Peening surface of the overlap weld; (**b**) Close-up image of the peening area.

**Figure 6 materials-16-00463-f006:**
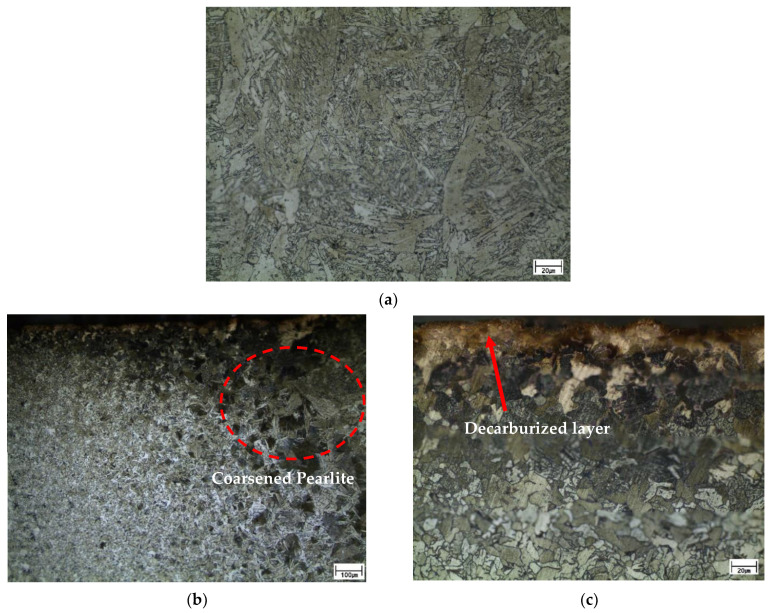
Microstructures of overlap-defect specimen without peening. (**a**) Center of the bead; (**b**) Heat-affected zone (HAZ); (**c**) Upper part of HAZ. (**b**) Low- and (**a**,**c**) high-magnification.

**Figure 7 materials-16-00463-f007:**
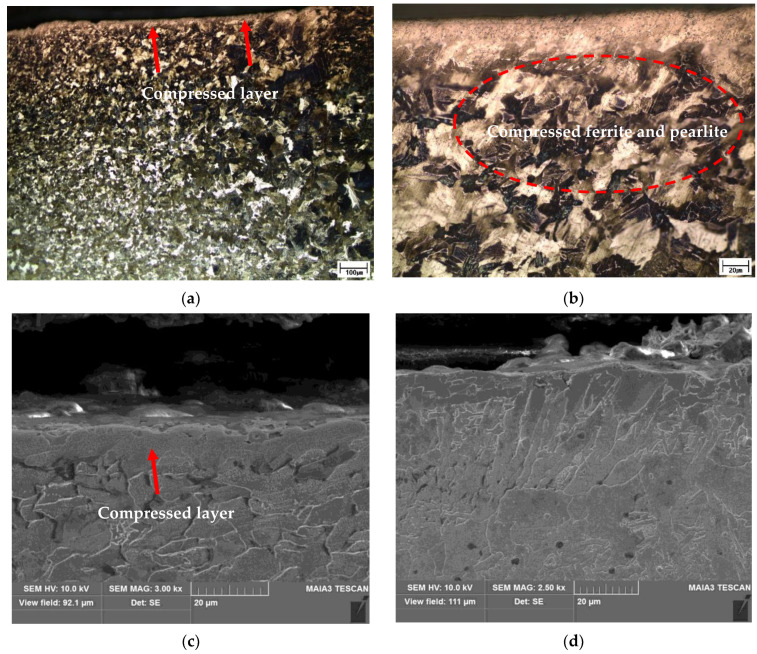
Microstructure of peened overlap-defect specimen. (**a**) Low-magnification optical image of the HAZ; (**b**) High-magnification optical image of the HAZ; (**c**) Scanning electron microscopy (SEM) image of the HAZ; (**d**) SEM image of the weld bead.

**Figure 8 materials-16-00463-f008:**
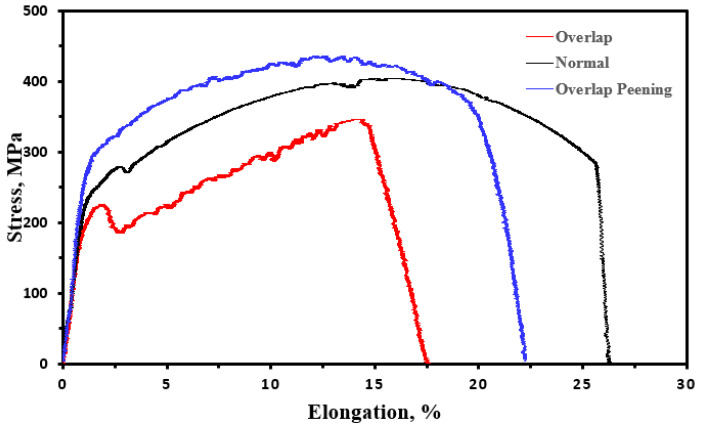
Stress–strain curves of welded specimens with a normal weld, overlap defect, and overlap defect after peening.

**Figure 9 materials-16-00463-f009:**
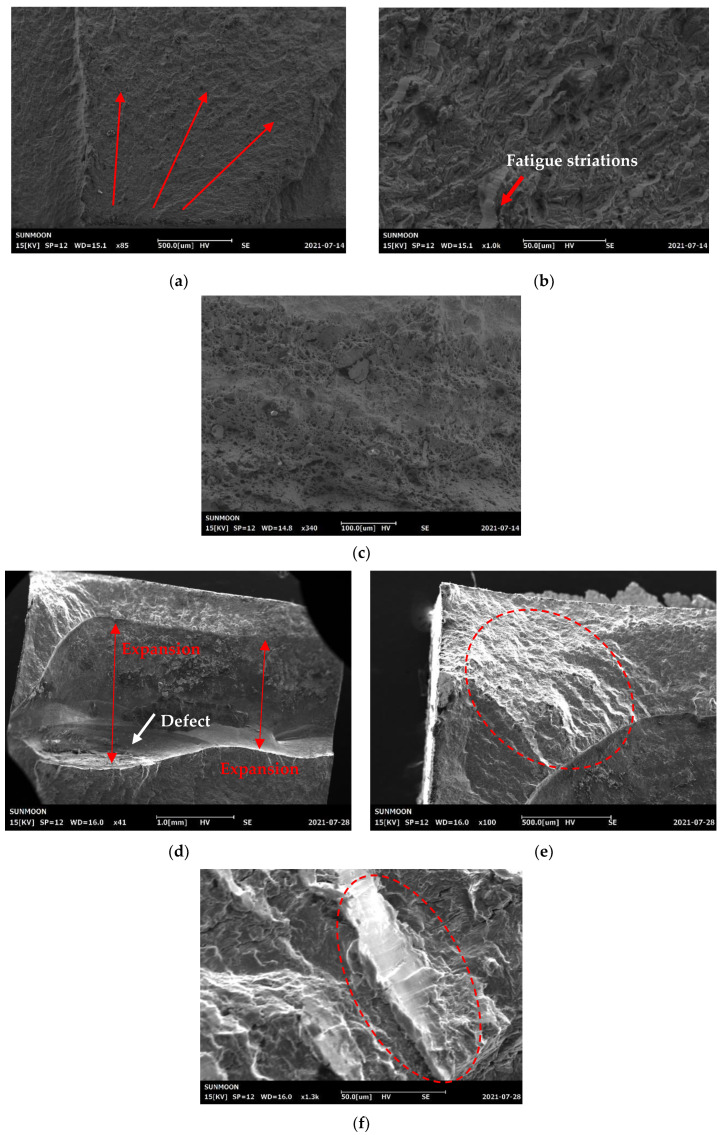
SEM images of the fracture surface of the normal welded specimen. (**a**) Low-magnification image; (**b**) High-magnification image; (**c**) High-magnification image of final fracture area. SEM images of the fracture surface of the overlap-defect welded specimens. (**d**) Low-magnification image; (**e**) Fatigue striation region; (**f**) High-magnification image of fatigue striations.

**Figure 10 materials-16-00463-f010:**
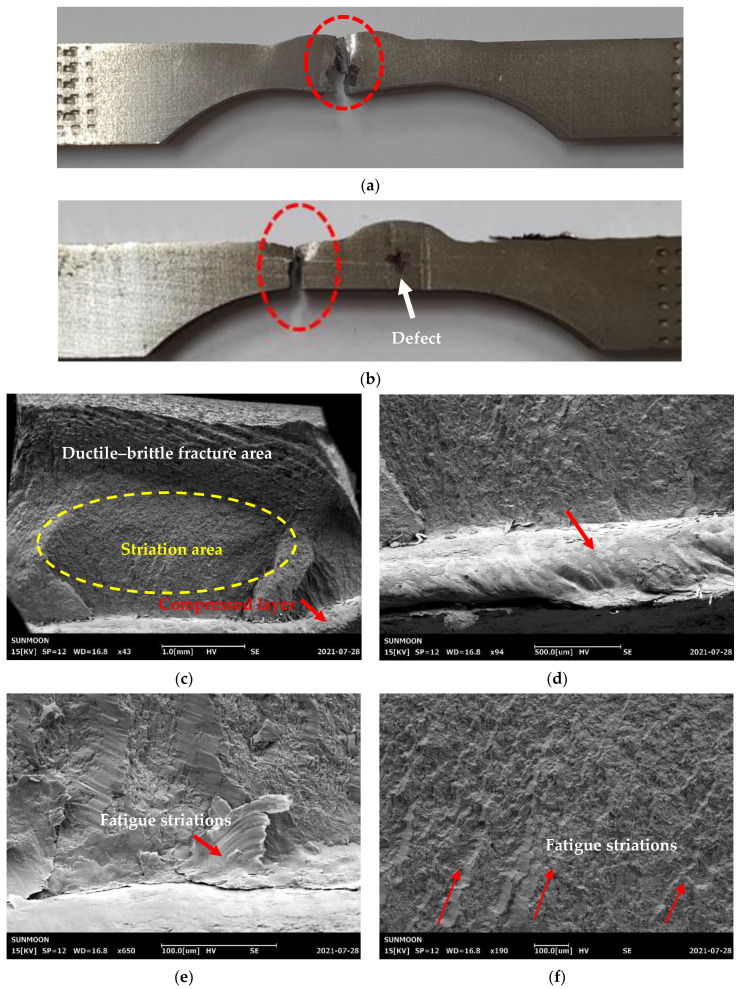
Photographs of the fatigue-fractured overlap-defect specimens. (**a**) Without peening; (**b**) With peening. SEM images of the fracture surface of the peened overlap-defect specimen. (**c**) Overall fracture surface; (**d**) Compressed surface; (**e**) Fatigue striations; (**f**) Progress of crack growth; (**g**) Ductile–brittle surface; (**h**) High-magnification image of the final fracture region.

**Figure 11 materials-16-00463-f011:**
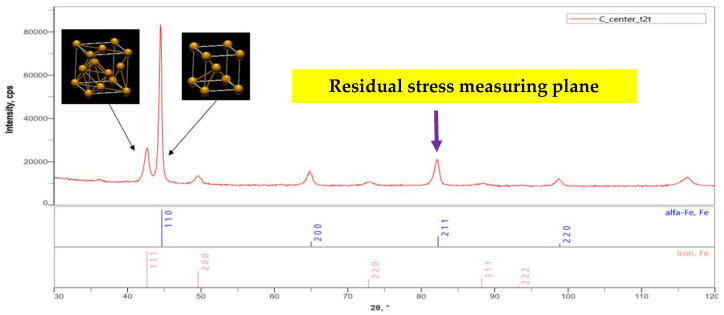
X-ray diffraction patterns of the normal welded specimen.

**Figure 12 materials-16-00463-f012:**
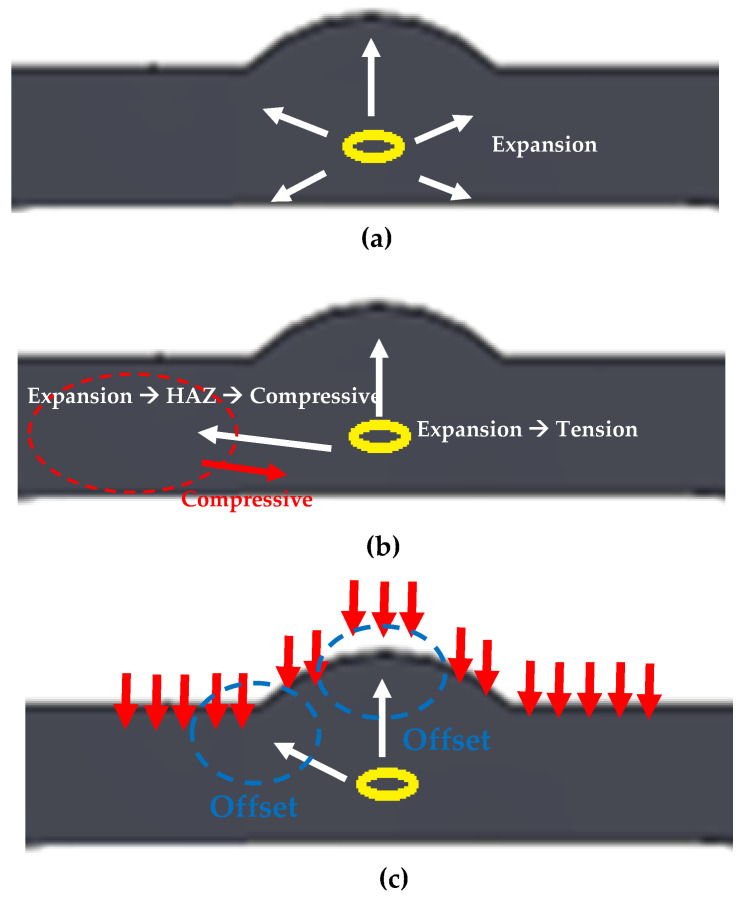
Schematic of residual stress generation at overlap defects. (**a**) Expansion force generated by the defect inside the bead; (**b**) Residual stress in the overlap-defect specimen; (**c**) Residual stress on the peened surface of the overlap-defect specimen.

**Figure 13 materials-16-00463-f013:**
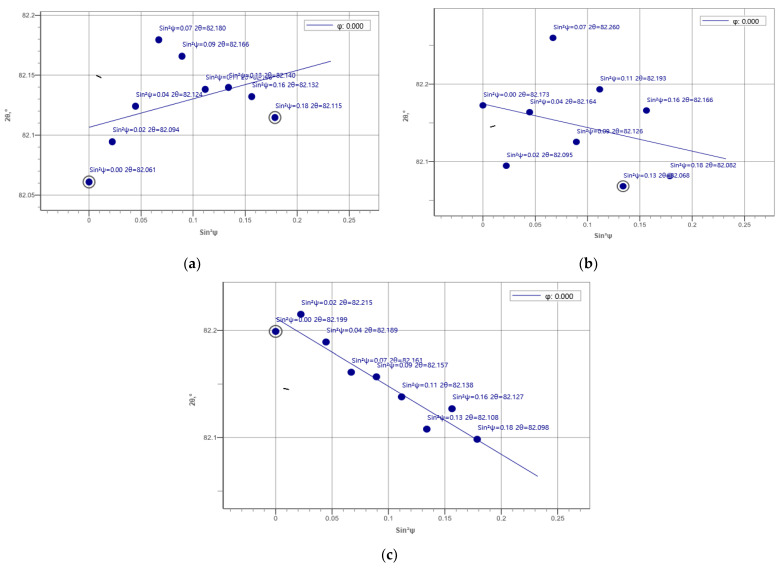
Residual stress diagrams of the overlap-defect specimen. (**a**) Heat-affected zone; (**b**) Upper region of the bead; (**c**) Middle region of the bead.

**Figure 14 materials-16-00463-f014:**
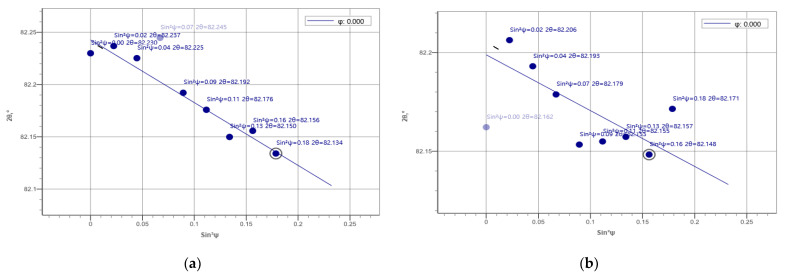
Residual stress diagrams of the peened overlap-defect specimen. (**a**) Heat-affected zone; (**b**) Upper region of the bead; (**c**) Middle region of the bead.

**Table 1 materials-16-00463-t001:** Microhardness of welded specimens at different depths and locations.

Depth from Surface (mm)	Hardness before Peening (HV)			Hardness after Peening (HV)		
	HAZ(L)	Bead	HAZ(R)	HAZ(L)	Bead	HAZ(R)
0.2	124.4	158.2	135.2	170.5	189.3	172.3
0.5	122.3	152.5	128.4	168.7	197.2	170.2
0.8	125.4	160.3	122.5	169.2	185.2	168.8
1.0	121.0	165.7	122.9	157.3	172.6	158.5
1.3	123.2	163.0	123.6	145.2	169.5	143.8
1.7	124.6	163.6	126.3	137.5	161.2	135.2
2.0	122.5	161.2	124.8	128.7	162.5	130.5
2.3	125.1	160.9	125.9	122.5	160.8	125.7
2.7	122.4	162.5	123.7	121.3	161.3	122.8
3.0	125.7	161.5	123.2	122.9	161.5	122.5
Average	123.7	160.9	126.6	144.4	172.1	145.0

**Table 2 materials-16-00463-t002:** Tensile test results of each specimen.

Property	Normal Specimen	Overlap-Defect Specimen	Peened Overlap-Defect Specimen
Yield strength (MPa)	277	231	289
Tensile strength (MPa)	399	345	434
Elongation (%)	26.5	17.8	22.1

**Table 3 materials-16-00463-t003:** Fatigue test results: number of cycles to failure.

Test Number	Normal Specimen	Overlap-Defect Specimen	Peened Overlap-Defect Specimen
1st	108,508	2543	121,012
2nd	118,210	2266	118,513
3rd	111,102	2439	119,322
4th	107,589	2355	120,583
5th	113,218	2512	115,420
Average	111,725	2423	118,970

**Table 4 materials-16-00463-t004:** Residual stress measurements of overlap-defect and peened test pieces.

Specimen	Location		
	Heat-Affected Zone	Upper Region of the Bead	Middle Region of the Bead
Overlap-defect specimen	−386 MPa	1496 MPa	1037 MPa
Peened overlap-defect specimen	978 MPa	460 MPa	647 MPa

## Data Availability

Not applicable.

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
