# Peer review of "Fatigue Life Improvement of Weld Beads with Overlap Defects Using Ultrasonic Peening"

_materials, 2023, doi:10.3390/ma16010463_

Round 1

Reviewer 1 Report

This paper elaborates ultrasound peening for correcting overlap defects. This is an interesting topic. However, the paper needs to be further revised.

1.     The literature review on ultrasonic peening and welding defects should be further added.

2.     The innovations and contributions of this paper need to be clearly elaborated.

3.     It is suggested that the main work and results of this paper be described in the Conclusion.

Author Response

We wish to re-submit the manuscript titled “Fatigue Life Improvement of Weld Beads with Overlap Defects Using Ultrasonic Peening.” The manuscript ID is materials-2117033. Details are written in the attached file. Thank you.

Reviewer 2 Report

Encouragement to the authors for presenting this very valuable research that also has practical results. I am very happy to read this manuscript which the subject is 100% in line with my experience and expertise. In my opinion, this manuscript can be published in the present form. But, to improve the quality of the manuscript, it is better to revised it using the following points:

1- The literature review should be improved in this field (i.e., fatigue and static strength improvement of welded parts using different surface treatment). However, the current introduction is about welding defects, inspections, repair and peening process. but it is necessary to discuss about other same research.

2-Related to Figure 3, it is better to state the used standard for tensile and fatigue test specimens. Also, are these two testing specimens the same geometry and dimensions? 

3- Related To table 2, it showed that the yield strength of normal stress is less than overlap-defect specimen, But in Figure 8 it is not the same results. Please check the results and correct it. 

4- Quality of Figure 8 is not proper and the writing is not clear. please, replace it with high quality picture. 

5- Related to Table 4, three different location were selected and residual stress values were reported, it is strongly suggested to display a residual stress diagram in terms of distance, which start from one point and move to other point and includes three desired location. In this case, the reader can see the residual stress variations in this type of connection. 

Author Response

(The authors gave the same response as above.)
